# DTTrans: PV Power Forecasting Using Delaunay Triangulation and TransGRU

**DOI:** 10.3390/s23010144

**Published:** 2022-12-23

**Authors:** Keunju Song, Jaeik Jeong, Jong-Hee Moon, Seong-Chul Kwon, Hongseok Kim

**Affiliations:** 1Department of Electronic Engineering, Sogang University, Seoul 04107, Republic of Korea; 2Smart Power Distribution Laboratory, KEPCO Research Institute, Daejeon 34056, Republic of Korea

**Keywords:** Delaunay triangulation, interpretable AI, TransGRU

## Abstract

In an era of high penetration of renewable energy, accurate photovoltaic (PV) power forecasting is crucial for balancing and scheduling power systems. However, PV power output has uncertainty since it depends on stochastic weather conditions. In this paper, we propose a novel short-term PV forecasting technique using Delaunay triangulation, of which the vertices are three weather stations that enclose a target PV site. By leveraging a Transformer encoder and gated recurrent unit (GRU), the proposed TransGRU model is robust against weather forecast error as it learns feature representation from weather data. We construct a framework based on Delaunay triangulation and TransGRU and verify that the proposed framework shows a 7–15% improvement compared to other state-of-the-art methods in terms of the normalized mean absolute error. Moreover, we investigate the effect of PV aggregation for virtual power plants where errors can be compensated across PV sites. Our framework demonstrates 41–60% improvement when PV sites are aggregated and achieves as low as 3–4% of forecasting error on average.

## 1. Introduction

Recently, the world has been converting fossil fuels-based primary energy resources to renewable energy to reduce carbon emissions and prevent climate change. Among various renewable energy sources, photovoltaic (PV) power generation has advantages in terms of noise and vibration compared to wind power. In addition, PV has a long lifespan and requires few maintenance costs. In 2021, the global total installed PV capacity was 167.8 GW, representing a 36% growth in one year [1]. However, PV power output is affected by the stochastic nature of weather conditions such as temperature, humidity, wind speed, and cloudiness, so it has uncertainty problems. Considering this inherent property, various solar PV systems have been designed to provide stable power quality for grid-integrated distributed resources systems [2,3,4]. Furthermore, in operating smart grid systems, accurate power generation forecasting of distributed energy resources has been required for scheduling and balancing power systems.

In the literature, many approaches have been suggested to improve forecast accuracy. Depending on the forecasting horizon, models can be categorized into three classes: short-term, medium-term, and long-term forecasting [5]. Short-term forecasting is performed on an hourly or daily basis and used for load balancing. Medium-term forecasting is conducted on a one-week or one-month basis and used for providing an estimation of power generation and demand. Long-term forecasting is usually carried out monthly or yearly. The aim of long-term forecasting is to plan PV installation.

In order to predict the PV power for the three classes above, various forecasting techniques have been proposed. Traditionally, physical models for solar energy prediction have been developed based on specific conditions of regions such as meteorological, climatological, and geographical parameters [6]. In addition, statistical models such as the auto-regressive integrated moving average (ARIMA) and the auto-regressive moving average with exogenous inputs (ARMAX) models have been applied to predict PV power generation [7,8]. However, physical models can only estimate solar energy for a longer time period, e.g., monthly, so they are hard to predict in detailed time intervals such as hours and minutes. Moreover, meteorological factors have a nonlinear relationship with PV power generation, so physical models and statistical models may not perform well. Recently, machine-learning-based models have been widely used since they can solve nonlinear problems well and outperform linear techniques. Artificial neural networks (ANN) [9,10], recurrent neural networks (RNN) such as long short-term memory (LSTM), GRU [11,12,13] and convolutional neural networks (CNN) [14,15] are the typical models. Furthermore, hybrid models also have been considered, such as CNN-LSTM and wavelet packet decomposition-LSTM (WPD-LSTM) [16,17].

Machine-learning-based PV power forecasting methods can be summarized as follows by the feature data that are related to PV power generation: (1) meteorological factors and historical PV power generation, (2) spatio-temporal features and (3) satellite or ground-based sky images. First, in [12], the authors used temperature, dew point, humidity and wind speed as input data to predict solar irradiance by LSTM. Moreover, the work in [13] additionally considered historical PV power. They used solar radiation, temperature, relative humidity, wind speed and historical PV power as an input of the CNN-bidirectional GRU (BiGRU) model to improve the performance. Secondly, several researchers considered spatio-temporal features, which are extracted from meteorological factors and geomorphology of the PV site. The work in [18] additionally used a digital elevation map of the solar PV power plant, solar irradiation, temperature, precipitation and wind speed to extract spatio-temporal features using CNN. Without using meteorological factors, the authors in [15] used the location of multiple PV sites to construct a space–time matrix with only historical PV power data and placed it as an input of CNN to extract spatio-temporal features. Thirdly, satellite and ground-based sky images are other approaches for extracting spatio-temporal features of PV power generation. In [19], the authors suggested two-stage predictions: the first is about cloud amount prediction by satellite images, and the second is about PV power prediction using the predicted cloud amount as one of the features. The authors in [20] used cloud images from ground-based cameras and historical solar irradiance for the input of the CNN model to predict solar irradiance by extracting spatio-temporal features. Recently, graph neural networks (GNN) have been considered to exploit spatio-temporal correlations in PV power generation, and further studies are still ongoing [21,22,23]. The work in [21] used the PV generation as signals on a graph to infer the part of cloud dynamics. In [23], the authors focused on interconnections among meteorological factors by factor-based graph modeling to capture complex weather variations. However, all the methods mentioned so far [9,10,11,12,13,14,15,16,17,18,19,20,21,22,23] have high-model complexity and thus may not be applicable to newly installed PV sites that do not have enough historical data. Considering that many new PV sites are expected to be installed, developing a light model is crucial to meet industrial needs.

In this regard, we propose a novel framework of day-ahead PV power forecasting using public weather station data in two aspects: weather station selection and weather data transformation. Since sufficient historical PV data or expensive satellite images are not available, we leverage the publicly available weather station data as follows. First, we construct Voronoi cells and Delaunay triangles based on the locations of weather stations. From these constructed regions, PV sites can exploit three nearby weather stations, which are the vertices of the Delaunay triangle that covers the target PV site. Second, we use a Transformer encoder to put more weight on seemingly important weather factors. The Transformer encoder shows high predictive performance and robustness regarding weather forecast error; this is possible because it transforms noisy input data to more useful time domain features, which are then fed into an RNN-based time series forecasting block. The proposed method is specifically useful for newly built PV sites because our model does not require the past PV generation data in the input of the model. For input data, our model uses four meteorological factors: temperature, wind, humidity, and cloud, which are relevant factors to PV power generation and collected far from the PV site, i.e., three vertices of the Delaunay triangle. Many other studies use high-quality meteorological factors such as solar radiation at the land surface and net solar radiation at the top of the atmosphere, collected from a local PV site [24]. Furthermore, some experiments assume that the weather forecast is correct, so the models under this assumption use historical weather data in the test phase [25]. Unlike this naive assumption, we use weather forecast data with various forecast error cases considering real cases. We summarize our key contributions as follows.

We develop a framework for PV generation forecasting by using Delaunay triangulation. Unlike the previous methods that use the weather data from the closest weather station, our method can select three weather stations that enclose the target PV site. We will see that this Delaunay triangulation-based approach outperforms selecting the three-nearest weather stations when the target PV site is well positioned in the corresponding Delaunay triangle.An interpretable AI model for PV power forecasting has not been actively studied so far. Considering this, we design an interpretable AI model called TransGRU, which is robust against weather forecast errors, by using a Transformer encoder. TransGRU learns input data through the Transformer encoder, which builds the feature importance for PV power generation in terms of feature representation learning. From the transformed feature data, our model can selectively exploit the weather stations and meteorological factors that are highly related to the target PV site.Our framework is simple but effective, specifically for newly built PV sites having a short period of data, e.g., one and a half years. We show that our framework overcomes the dependency on historical PV data by using Delaunay triangulation and a Transformer encoder. Specifically, we provide extensive experiments using 1034 PV sites nationwide and 86 weather stations in Korea. The results show that our framework achieves a 7–15% improvement in forecasting individual PV power generation and a 41–60% improvement in PV aggregation in forming a virtual power plant (VPP). As a result, we achieved a 3–4% of forecast error for VPP, which sufficiently satisfies the requirement of 6% or less error to participate in the renewable energy wholesale market run by the Korea Power Exchange (KPX).

The rest of this paper is organized as follows. In Section 2, we describe the methodologies and propose the TransGRU-based forecasting technique. In Section 3, we describe the model selection of TransGRU. The experimental results are provided in Section 4, followed by the conclusion in Section 5.

## 2. Proposed Methodologies

### 2.1. Overall Framework

Figure 1 shows the overall process of the proposed framework called DTTrans and its three steps: data preprocessing, training phase, and test phase. In the first step, PV sites are clustered based on the Delaunay triangle to identify which weather stations enclose each of them. After that, a union set of identified weather stations is fed into a rule-based selection algorithm called Union-Inner Triangles in order to select three highly relevant weather stations in an adaptive way. This process extends the vanilla Delaunay Triangulation to DT+, as shown in Figure 1. The dataset is then split into a training set, validation set, and test set. In the second step, the proposed TransGRU model is trained by the training set, and the hyperparameters of TransGRU are determined by the validation set. In the final step, we estimate the performance for the test set for various cases.

### 2.2. Voronoi Tessellation and Delaunay Triangulation

Voronoi Tessellation and Delaunay Triangulation have been used in a practical and theoretical way in many fields, such as science, technology and visual art [26,27,28,29,30,31]. Given a set of seed points in the plane, their Voronoi Tessellation divides the plane according to the nearest-neighbor rule [32].

Let $p_{i}\in R^{2}$ denote the coordinates of a seed point, i.e., the *i*-th weather station in our case. Then, a set of *n* seed points, i.e., *n* weather stations, is denoted by
(1)$$P=\{p_{1},p_{2},\ldots ,p_{n}\}.$$

Then, the Voronoi cell of *i*-th weather station is given by
(2)$$V\left(p_{i}\right)=\{p\in R^{2}|\;||p-p_{i}\left|\right|\leq ||p-p_{j}\left||,\;\forall j\neq i\right\}$$
where ||·|| is a distance metric. Equation (Equation 2) is about how the Voronoi cell is constructed. Arbitrary point *p* and seed points $p_{i}$ are on the same plane, which are the locations of the PV power plant (subspace of *p*) and weather station in our case. Under this status, the distance between *p* and seed point (e.g., $p_{i}$) is calculated and compared with other seed points to find which seed point is closest. Hence, $p_{i}$ serves as the *closest* weather station to any points in $V(p_{i})$. A set of Voronoi cells is called the Voronoi Tessellation (VT) (or Voronoi diagram) of P and denoted by:(3)$$V\left(P\right)=\{V\left(p_{1}\right),V\left(p_{2}\right),\ldots ,V\left(p_{n}\right)\}.$$

The Delaunay tessellation is considered to be dual to VT [33]. Given V(P), connecting all pairs of the seed points whose Voronoi cells share an edge constructs the Delaunay tessellation. Let $\bar{p_{i}p_{j}}$ be the line connecting $p_{i}$ and $p_{j}$ and $e(p_{i},p_{j})$ be an edge between the Voronoi cells $V(p_{i})$ and $V(p_{j})$. Then, in the Delaunay tessellation of P, denoted by D(P), there exists an edge $\bar{p_{i}p_{j}}$ if and only if $e\left(p_{i},p_{j}\right)\in V\left(P\right)\neq \emptyset$. If $e(p_{i},p_{j})\neq \emptyset$, then $V(p_{i})$ and $V(p_{j})$ are considered adjacent. If the Delaunay tessellation consists of only triangles, it is called Delaunay Triangulation (DT). VT and DT are not only dual graphs but also mutual figures [33]. In this way, we can construct V(P) and D(P), which are mutually the nearest regions of *P*, i.e., the set of weather stations. As an example, Figure 2 shows the VT and DT constructed from the 86 nationwide weather stations in Korea.

### 2.3. Union-Inner Triangles Algorithm

Now we present how to select relevant weather stations given a target PV site. One weather station can be used for PV forecasting if the PV site is in the corresponding Voronoi cell. This is a typical method used so far, i.e., choosing the nearest weather station. On the other hand, if the PV site is in the corresponding Delaunay triangle, three weather stations at the vertices of the triangle, called DT weather stations hereafter, can be used for PV forecasting. However, using DT weather stations may not be good depending on the position of the PV site as well as the shape of DT, specifically when DT is a very flat triangle, as can be seen in some regions of Figure 2b. In addition, DT weather stations can sometimes be very different from the three-nearest weather stations. Hence, we need to have a method that can discern whether to use DT weather stations or the three-nearest weather stations. The proposed algorithm considers the *union* set of DT weather stations and the three-nearest weather stations. The size of this union set can be either 3, 4, 5 or 6. When the size of the union set is 3, it implies that DT weather stations are also the three-nearest weather stations, which is an ideal case. By contrast, as can be seen in Figure 3, the size can be 4 or larger, depending on the shape of DT and the inside location of the PV site. To investigate this, we focus on three *inner* triangles in the DT and their interior angles; an obtuse DT is not favorable, nor are the obtuse inner triangles. Furthermore, when the PV site is close to a side, either of the three inner triangles becomes a severe obtuse triangle. To simply capture this, we want the interior angles, denoted by $a_{1},b_{1},a_{2},b_{2},a_{3},b_{3}$ in Figure 3a, to not be less than some threshold; otherwise, the three-nearest weather stations are selected. This algorithm has the limitation that it is a heuristic way to select three weather stations; however, it can be the one of examples that captures the most relevant weather stations for PV power forecasting. The pseudo code of the proposed algorithm is summarized in Algorithm 1, and various cases are shown in Figure 3b.


**Algorithm 1** Union-Inner Triangles Algorithm


U←{{verticesofDelaunaytriangle}⋃{3nearestweatherstations}}



$\left|U\right|∼\left\{3,4,5,6\right\}$

$a_{1},b_{1},a_{2},b_{2},a_{3},b_{3}\leftarrow$ the angles of 3 inner triangles$\epsilon_{1},\epsilon_{2}\leftarrow$ the threshold angles**if** 
$\left|U\right|=3$
**then**    Use DT weather stations**else if** 
$\left|U\right|=4$
**then**    **if** $\min\left\{a_{1},b_{1},a_{2},b_{2},a_{3},b_{3}\right\}<\epsilon_{1}$ **then**        Use 3 nearest weather stations    **else**        Use DT weather stations    **end if****else if** 
$\left|U\right|=5$
**then**    **if** $\min\left\{a_{1},b_{1},a_{2},b_{2},a_{3},b_{3}\right\}<\epsilon_{2}$ **then**        Use 3 nearest weather stations    **else**        Use DT weather stations    **end if****else if** 
$\left|U\right|=6$
**then**    Use 3 nearest weather stations
**end if**




### 2.4. Proposed TransGRU Model

We now make vectors x and y, which are the input and output of our TransGRU model. Let $x,y\in R^{time}$ represent the hourly time-series values. x has 4 meteorological factors (temperature, wind speed, humidity, and cloud) per weather station during a day. By contrast, y has hourly PV generation during a day. Figure 4 shows the form of the input vector x based on VT and DT.

In doing this, we leverage the Transformer encoder, which is then concatenated by GRU for time series forecasting. Details are as follows.

#### 2.4.1. Transformer Encoder

The Transformer encoder is a part of the Transformer, which is a well-known model in machine translation [34]. In the Transformer encoder, as shown in Figure 5, multi-head attention is the main technique. Instead of a single attention function with queries, keys, and values, multi-head attention linearly projects the queries, keys and values. Furthermore, multi-head attention allows the model to attend to different representations at different positions from the number of heads, denoted by *h*. We use a single Transformer encoder layer and a single head (i.e., h=1), which is single self-attention. Thus, we have multi-head attention as follows by using the same notations in [35]:(4)$$\begin{array}{c} MultiHead\left(Q,K,V\right)=Concat\left(head\right)W^{O}\\ where\;head=softmax(QW^{Q},KW^{K},VW^{V}) \end{array}$$
where the projections are parameter matrices $W^{Q}\;\in \;R^{d_{model}\times d_{k}},W^{K}\;\in \;R^{d_{model}\times d_{k}},W^{V}\;\in \;R^{d_{model}\times d_{v}}$ and $W^{O}\;\in \;R^{hd_{v}\times d_{model}}$. The variables $d_{k}$, $d_{v}$ and $d_{model}$ denote the dimension of keys, values and outputs of the Transformer encoder. We use $d_{k}=d_{v}=d_{model}/h=96$ for PV sites in the Voronoi cell and $d_{k}=d_{v}=d_{model}/h=288$ for PV sites in the Delaunay triangle. Applying a single Transformer encoder with single a self-attention mechanism allows the model to selectively focus on input features that are more relevant to the current output and alleviate the intervention of other features.

#### 2.4.2. Gated Recurrent Unit (GRU)

Gated recurrent unit (GRU) is a well-known RNN model, which is slightly different from a long short-term memory network (LSTM). It uses reset gate and update gate to solve the long-term dependencies of different time scales and takes the sum between the newly computed state of the data and the existing hidden state to solve the gradient vanishing problem. GRU has fewer parameters than LSTM since it has only two gates. In [36], the authors showed that GRU has fast convergence compared to LSTM while maintaining the performance of LSTM in certain tasks, depending on a dataset. We also find that the performance of GRU and LSTM is similar, so either can be a flexible choice for recurrent neural networks, and we choose GRU considering model complexity. Figure 6 shows the architecture of the GRU unit. The relationship between the input and output of GRU can be described as follows:(5)$$\left\{\begin{array}{c} r\left(t\right)=\sigma_{g}\left(W_{r}x\left(t\right)+U_{r}h\left(t-1\right)+b_{r}\right)\\ z\left(t\right)=\sigma_{g}\left(W_{z}x\left(t\right)+U_{z}h\left(t-1\right)+b_{z}\right)\\ h\left(t\right)=\left(1-z\left(t\right)\right)⊙h\left(t-1\right)+z\left(t\right)⊙\hat{h}\left(t\right)\\ \hat{h}\left(t\right)=\sigma_{h}\left(W_{h}x\left(t\right)+U_{h}\left(r\left(t\right)⊙h\left(t-1\right)\right)+b_{h}\right) \end{array}\right.$$
where z(t) is the update gate vector at time step *t*, r(t) is the reset gate vector, *W*, *U* and *b* are parameter matrices and vector. The operator ⊙ denotes the Hadamard product. $\sigma_{g}$ is a sigmoid function, and $\sigma_{h}$ is a hyperbolic tangent.

Using the Transformer encoder and GRU, our proposed TransGRU model adaptively represents the input features that are highly related to the output and solves the long-term dependencies and the gradient vanishing problem by using represented feature data.

## 3. Model Selection

In this section, we describe PV generation data and weather data (i.e., input and output data of the model) used for training, validation and testing. Then we describe the hyperparameters of the proposed TransGRU, the compared models and the model optimization.

### 3.1. Data Description

We used the PV generation data of Korea provided by the Korea Electric Power Corporation (KEPCO). There are 1034 sites located in all parts of the country, and PV generation data from newly built sites are from 1 December 2019 to 28 October 2021 every hour. Table 1 shows the statistical information (maximum, mean, conditional mean, i.e., during effective daytime, and standard deviation) of sampled PV data from each location. The data are normalized between 0 and 1 using the installed PV capacity. The dataset is split into training set (80%), validation set (10%), and test set (10%).

We used Automated Synoptic Observing System (ASOS) weather data of Korea released by the Korea Meteorological Administration (KMA) [37]. There are n=86 weather stations located in all parts of the country, and weather data were obtained for the same period of PV generation. ASOS weather data provides 16 meteorological factors every hour, and we selected 4 meteorological factors: temperature, wind speed, humidity and cloud amount. The linear interpolation method is used to fill the missing values after data cleaning. In addition, the data are normalized using the maximum value of each factor.

### 3.2. Hyperparameters of the Proposed TransGRU

The size of the input vector is determined by the number of time steps per day. As shown in Figure 4, the size of time steps depends on the number of selected weather stations. We determine time=96 when the PV site is clustered in the Voronoi cell, and time=288 when the PV site is clustered in the Delaunay triangle. For example, the input vector for DT is formed as $x\in R^{288}$. For the output vector, the size of time steps is fixed by 24 for hourly PV generation during a day. Therefore, the shape of the output vector becomes $y\in R^{24}$.

In Section 2.4.2, we had provided the reason for the choice of RNN models; LSTM and GRU. We already performed experiments about using LSTM; however, it was found that the performance of LSTM and GRU are similar. Therefore, either can be a flexible choice, but we chose GRU considering the model complexity and generalization; GRU has a smaller number of model parameters than LSTM. The reason for only comparing LSTM with GRU is that these methods are the most well-known and the basis of many other RNN models. Then we determined the hyperparameters of each component in TransGRU, i.e., Transformer encoder and GRU using a validation set. For the Transformer encoder, we selected 1 head for multi-head attention, which is equal to single self-attention. We set 512 as the dimension of a single hidden layer in the feed-forward network and 1 encoder layer. For GRU, we selected the single GRU unit and set the size of the hidden state as 64. Two fully connected (FC) layers are followed behind, and we set 256 as the dimension of each layer. For an activation function, ReLU [38] is adopted in a feed-forward network of a Transformer encoder and two fully connected layers. The structure of TransGRU is shown in Table 2.

### 3.3. Hyperparameters of the Compared Models

We compare the proposed TransGRU model with two well-known models: multi-layer perceptrons (MLP) and GRU [39]. MLP was chosen as a baseline because complex models suffer from overfitting with insufficient training data from newly built PV sites. GRU was chosen over LSTM because they have similar performance, but GRU has lower model complexity. The MLP-based model uses 3 hidden layers, and each of them consists of 256 units. For an activation function, ReLU [38] is adopted. The GRU-based model uses a single GRU unit and two fully connected layers. The size of the hidden state is 64, and the number of dimensions of two layers is 256, which is the same as the last two components of the TransGRU model.

### 3.4. Hyperparameters of the Model Optimization

All models were trained using the hyperparameters in Table 3. We used the Adam optimizer [40] with a learning rate of 0.0001. The batch size was selected as 18 for 1000 epochs. Lastly, we selected the mean squared error (MSE) as our objective function.

By setting the hyperparameters in this way, Figure 7 shows that our models are trained well on the dataset without overfitting and underfitting problems.

## 4. Experiment Results

In this section, we describe the performance metric, the reproduction of weather forecast data, performances of union-inner triangles, the transformed input data and the experimental results for individual and aggregated PV generation forecasting.

### 4.1. Performance Metric

In order to evaluate the performance of prediction models, we mainly use the normalized mean absolute error (NMAE10):(6)$$NMAE10\left(\%\right)=\frac{100}{T}\sum_{t=1}^{T}\left|y_{t}-{\hat{y}}_{t}\right|1_{\left(y_{t}\geq 0.1C\right)}$$
where $y_{t}$ is the ground truth PV generation, ${\hat{y}}_{t}$ is the predicted PV generation, *T* is the number of prediction intervals, *C* is the installed PV generation capacity and $1_{\left(\right)}$ is an indicator function. Since error is measured in KPX only when the PV generation is at least 0.1C, i.e., during the effective daytime, our performance metric accounts for this.

### 4.2. Reproduction of Weather Forecast Data

ASOS does not provide weather forecast data, only historical weather data. Hence, to be realistic, we reproduce weather forecast data in the test set by using additive white Gaussian noise (AWGN) [34], which is given by:(7)$$x\left(t\right)\leftarrow x\left(t\right)+N\left(0,\sigma^{2}\right)$$
where x(t) is ASOS weather data at time step *t*. To consider diverse weather forecast errors, we vary σ to the rate of 5%, 10%, 15% and 20% of the *maximum* value of each meteorological factor.

### 4.3. The Impact of Delaunay Triangulation and Union-Inner Triangles Algorithm

Recall that we proposed the union-inner triangles algorithm in Section 2.3 to extend the Delaunay triangle-based weather station selection; the target PV site can use either DT weather stations or the three-nearest weather stations. In the experiment, we found that DT weather stations and the three-nearest weather stations are the same for 348 PV sites out of 1034 PV sites, but different for the rest of the 686 PV sites. As shown in Figure 3, the number of PV sites for the size of the union set is determined as $\left|U\right|=3$ for 348 PV sites, $\left|U\right|=4$ for 528 PV sites, and $\left|U\right|=5$ for 158 PV sites. In our case, none of the PV sites have $\left|U\right|=6$. Now, it is interesting to see which one would be better for the rest of the 686 PV sites when $\left|U\right|>3$: DT weather stations or three-nearest weather stations. In doing this, Figure 8 shows the distribution of 686 PV sites. As can be seen, 472 PV sites (i.e., red triangle symbol) show that using DT weather stations performs better than the three-nearest weather stations. We found that DT is less effective when PV sites are located in obtuse triangles or close to the side of a triangle. Figure 8b shows an example when the three nearest neighbors are better than DT; we can infer that if PV sites are located in the way like green plus points in Figure 8b, which is the case that DT weather stations can be very different from the three-nearest weather stations, then using the nearest three weather stations can be better than using DT.

In the implementation of the union-inner triangles algorithm, we set the degree parameters to $\epsilon_{1}=1.3$ and $\epsilon_{2}=2$. The choice of $\epsilon_{1}$ and $\epsilon_{2}$ is based on validation results of a comparison between DT weather stations and the three nearest weather stations, which shows the pattern of the specific locations of PV sites. As an example, Figure 9 shows the CDFs of NMAE10 for one of the Delaunay triangles in Daegu and Gwangju, respectively. As can be seen, our proposed method, denoted by DT+, outperforms the three-nearest weather stations and Voronoi in both cases by effectively selecting three *relevant* weather stations in the preprocessing step.

### 4.4. The Impact of Transformer Encoder

Thus far, we have discussed the selection of relevant weather stations. Now we discuss the forecasting model using the weather data from those weather stations. As shown in Table 2, the Transformer encoder is located in the first and the second layer of the TransGRU model, and it transforms input data into the feature domain. In other words, TransGRU learns input data for better representation and feed-forwards it to the recurrent learning task. In making an interpretable model, we compared the raw input data and transformed input data of three selected weather stations A, B, and C, as shown in Figure 10.

As can be seen, the transformed data have high or low values at certain time steps, compared to the raw data in each section: temperature, wind speed, humidity and cloud amount. From the transformed data, we can specifically infer which meteorological factor is highly considered at a certain time step because the role of the Transformer encoder is to learn and transform the input data to the feature data with the same dimension. Furthermore, TransGRU considers the importance of weather stations adaptively, which may not be similar to the order of distance to the weather stations. For example, the averages of the transformed input data of Figure 10a–c are 0.46, 0.49, and 0.50, respectively. Hence, one may think that weather station C is the closest to the target PV site, and weather station A is the farthest away from the target PV site. However, as shown in Figure 11, in fact, weather station B is the farthest but has more effect than weather station A. Therefore, the Transformer encoder performs feature representation learning to make a subsequent learning task easier, which builds robustness for the weather forecast error [41]. The impact of the Transformer encoder becomes more evident when we see the performance improvement of TransGRU over GRU in Section 4.5.

### 4.5. Forecasting of Individual PV Sites

Figure 12 shows the CDF comparison of MLP, GRU and TransGRU in Gwangju and Daegu, respectively. As can be seen, the proposed TransGRU is located in the upper left for all weather forecast error cases. Furthermore, the performance gap between TransGRU and others becomes larger as the weather forecast error increases. In Figure 12b, as the *weather* forecast error goes from 5% to 20%, TransGRU further outperforms MLP and GRU. More specifically, for the 5% *weather* forecast error case, differences between TransGRU and MLP, GRU in terms of NMAE10 are 3.81% and 10.31%. For the 20% *weather* forecast error case, differences between TransGRU and MLP, GRU increase, which show 7.28% and 11.57%. In addition, the differences between the 5% and 20% *weather* forecast error cases for each MLP, GRU, and TransGRU itself are 12.72%, 10.20%, and 8.65%, which show TransGRU has the smallest decrease. This confirms that the Transformer encoder is robust for the weather forecast error. One may wonder why MLP is better than GRU. This is because GRU (or LSTM) requires more data to be effective, but, in our case, newly built PV sites do not have sufficient data.

In addition to CDF comparison, the averages of NMAE10 in Gwangju and Daegu are shown in Table 4. In addition, we provide the result using VT to highlight the performance improvement using DT+ over VT in Table 4. As described in the table, DT+ outperforms VT for all the cases of weather forecasting error and the choice of learning model. For example, DT+ shows 19.9%, 16.0%, and 10.3% improvement compared to VT in the 20% weather forecast error for Gwangju. In addition, TransGRU outperforms MLP and GRU as well. For example, TransGRU with DT+ shows 5.9% and 15.1% improvement compared to MLP and GRU with DT+ in the 20% weather forecast error for Gwangju. Therefore, it demonstrates that DT+-based TransGRU works well in PV power forecasting.

### 4.6. Forecasting of Aggregated PV Sites

Finally, we investigate the effect of PV site aggregation, which is the concept of VPP [42]. VPP is a cloud-based distributed power plant that aggregates the capacities of heterogeneous distributed energy resources (DER) for the purposes of enhancing power generation, as well as trading or selling power on the electricity market. Since aggregating PV sites can improve the forecasting accuracy substantially, it is of interest to see the VPP performance of the proposed DTTrans model. Power generation from individual PV sites may severely fluctuate due to stochastic weather conditions, but power generation becomes smoother as more PV sites are aggregated. Therefore, PV aggregation can achieve economical profit by stable PV power generation. In this case, PV power forecasting can be different from the individual PV sites case because forecasting errors from one site can be compensated by other sites. To implement this, we normalize the aggregated data between 0 and 1 using the sum of installed PV generation capacities. Table 5 shows the results of PV aggregation in Gwangju and Daegu, respectively. As can be seen, PV aggregation shows a 30−45% improvement compared to the individual PV site in Table 4.

Rather than simply aggregating PV sites colocated in some regions, one can consider the best combination of multiple PV sites based on meteorological satellite images, as done in [43]. We can seek a method to obtain the best combination of multiple PV sites, but since the proposed DTTrans is in parallel to constructing VPP, here we simply focus on randomized VPP construction; the randomized VPP may not be better than the best combination method, but it can capture the performance in the real field because VPP construction cannot always be performed at will. In Figure 13, we compare the CDFs of NMAE10 between the DT+-based VPP and individual PV sites. DT+-based VPP is constructed to meet the requirements in Korea; VPP capacity should be larger than 20MW, and capacities of each PV site should be lower than 1MW. In implementing this, we randomly select PV sites and aggregate them to make 200 VPPs. It shows that the DT+-based VPP substantially improves the performance from 8.7% to 4.2%. The CDF shown in Figure 13 clearly shows this improvement. Considering that the renewable energy wholesale market in Korea currently requires 6% or less forecasting error, the proposed method sufficiently satisfies this requirement.

## 5. Conclusions

In this paper, we proposed a novel short-term PV power forecasting technique called DTTrans based on DT and TransGRU. Unlike using weather data from the closest weather station, DT selects three weather stations that enclose PV sites. Since the effect of DT weather stations depends on the shape of DT and the inside location of the PV site, the proposed union-inner triangles algorithm adaptively handles whether to use DT weather stations or the three-nearest weather stations. The proposed method is then combined with an interpretable AI model called TransGRU and becomes robust against weather forecast error since it selectively considers the weather stations and meteorological factors that are highly related to PV generation. Extensive experiments show that the proposed DTTrans is highly effective in forecasting individual PV sites as well as aggregated PV sites. As the weather forecast error increases from 5% to 20%, TransGRU further outperforms the other methods based on MLP and GRU. Furthermore, DTTrans considerably improves the performance when multiple PV sites are aggregated in the VPP form and finally achieves a 3–4% forecasting error on average.

## Figures and Tables

**Figure 1 sensors-23-00144-f001:**
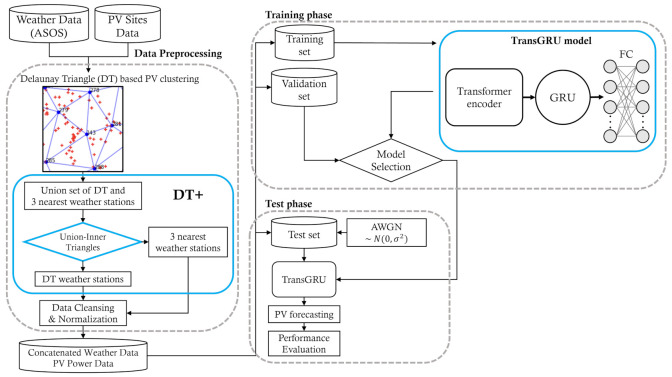
Overall process of the DTTrans framework.

**Figure 2 sensors-23-00144-f002:**
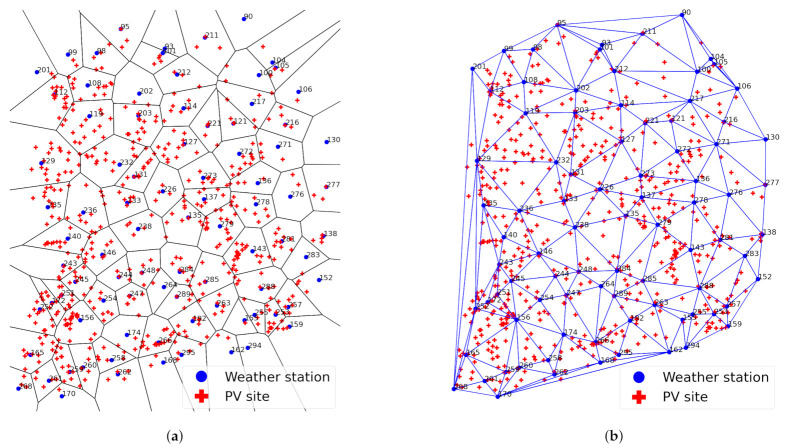
PV sites (red cross), Voronoi diagram and Delaunay triangulation of weather stations (blue circle) in Korea. (**a**) Voronoi diagram of weather stations. (**b**) Delaunay triangulation of weather stations.

**Figure 3 sensors-23-00144-f003:**
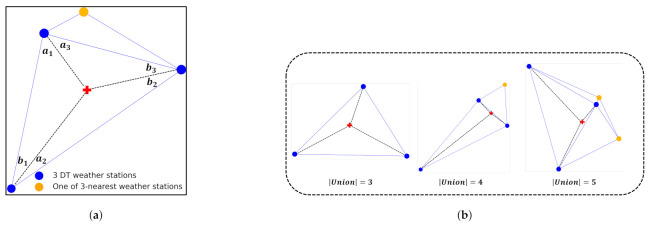
Extension of DT to DT+ using union-inner triangles algorithm. (**a**) Union-inner triangles. (**b**) Cases for the union set of DT weather stations and 3 nearest weather stations.

**Figure 4 sensors-23-00144-f004:**
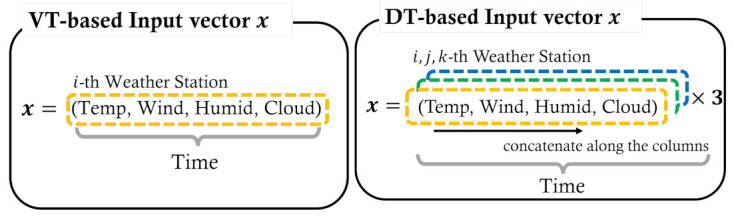
Input vector x for Voronoi Tessellation and Delaunay Triangulation.

**Figure 5 sensors-23-00144-f005:**
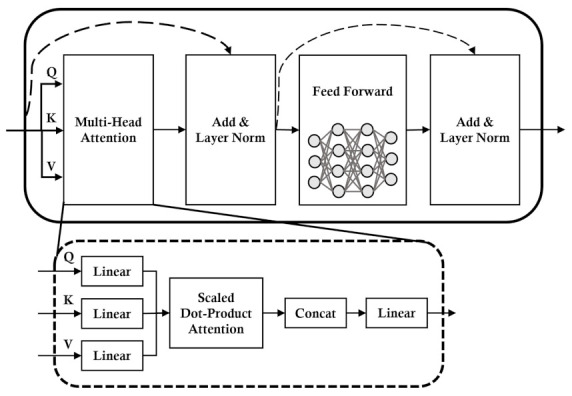
Architecture of Transformer encoder and multi-head attention.

**Figure 6 sensors-23-00144-f006:**
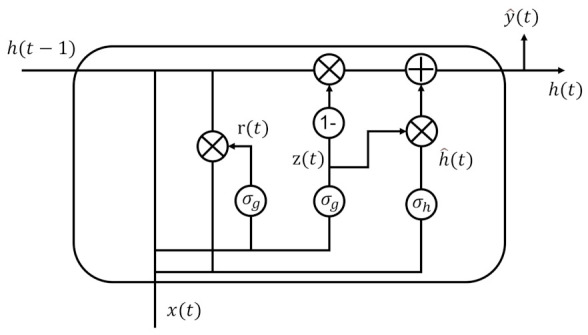
Architecture of the GRU unit.

**Figure 7 sensors-23-00144-f007:**
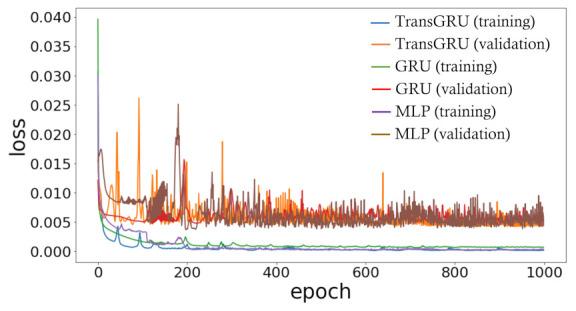
Learning curve comparison during training and validation.

**Figure 8 sensors-23-00144-f008:**
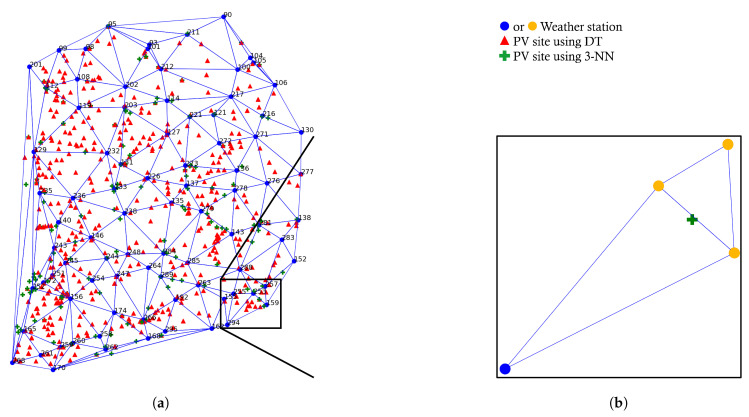
The result of union-inner triangles preprocessing. (**a**) When DT weather stations and 3-nearest weather stations are different, 66% of PV sites (red triangle) use DT weather stations and 34% of PV sites (green cross) use the 3 nearest weather stations. (**b**) An example when DT is not a good choice.

**Figure 9 sensors-23-00144-f009:**
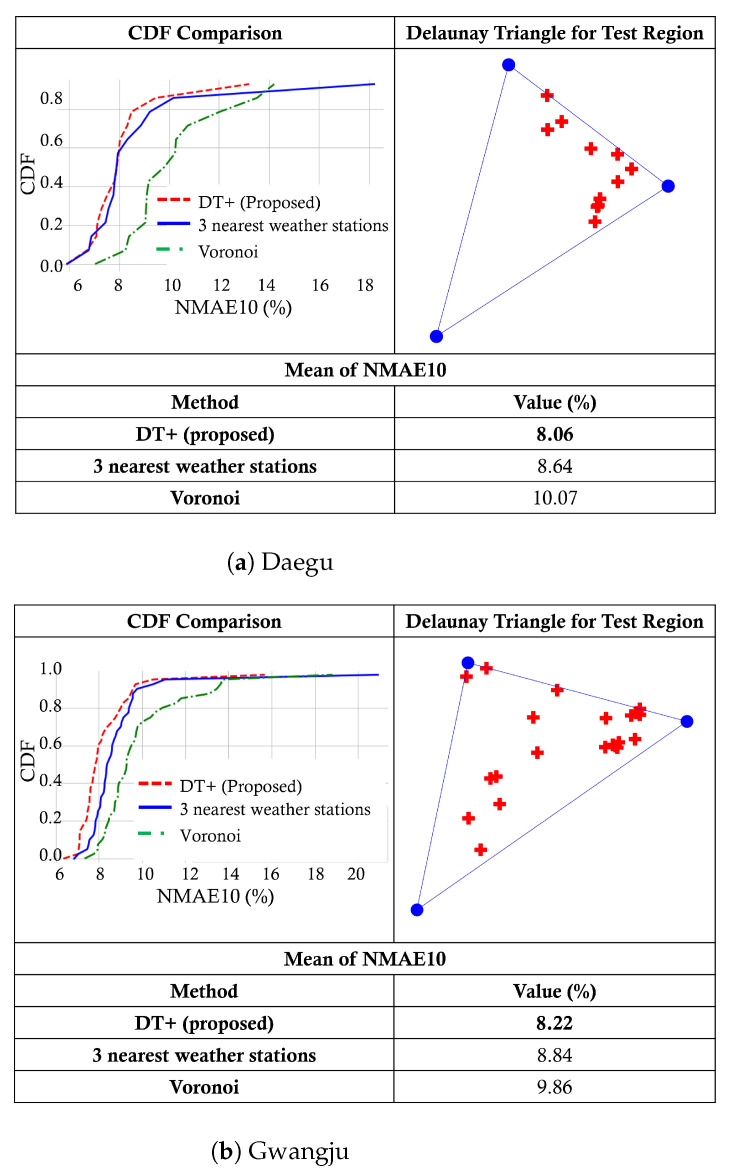
An example of union-inner triangles test results.

**Figure 10 sensors-23-00144-f010:**
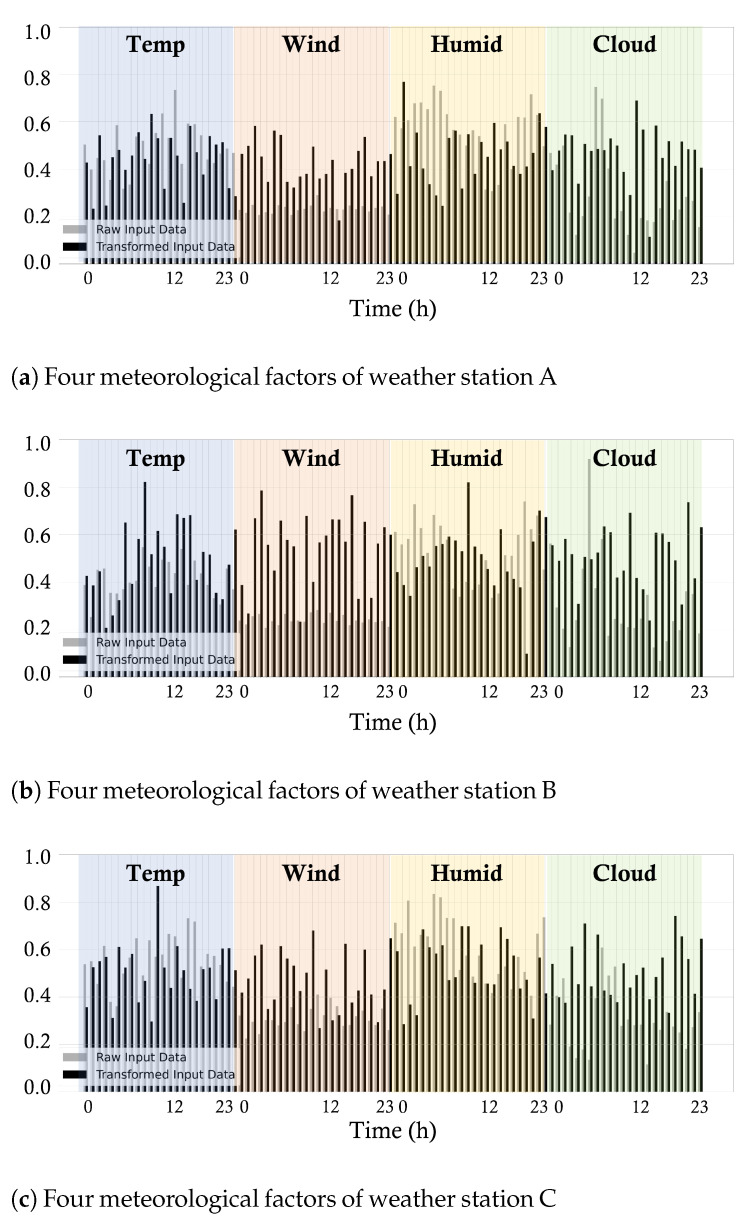
Comparison of raw input data and transformed input feature.

**Figure 11 sensors-23-00144-f011:**
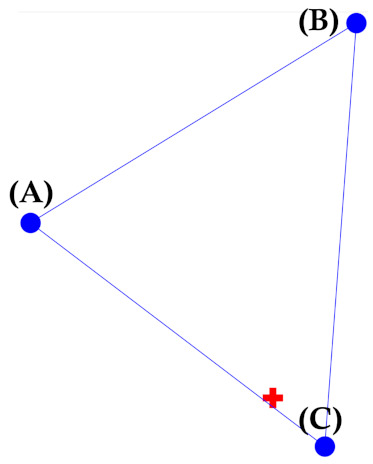
The location of 3 weather stations and the target PV site.

**Figure 12 sensors-23-00144-f012:**
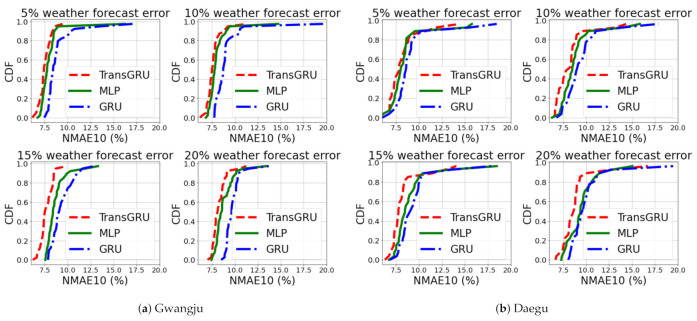
CDF comparison (individual PV sites forecasting).

**Figure 13 sensors-23-00144-f013:**
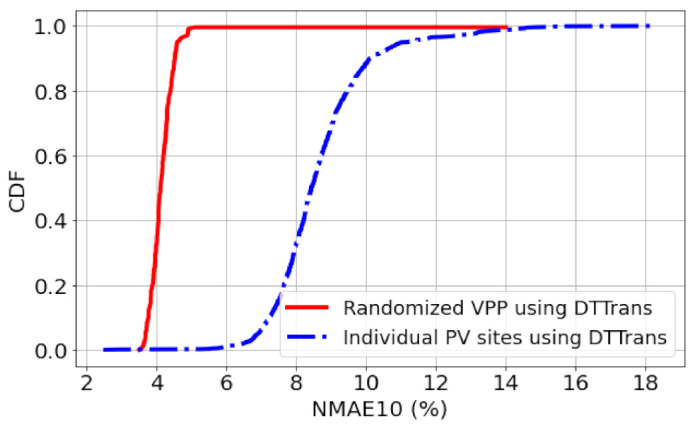
CDF comparison in terms of randomized VPP in Korea.

**Table 1 sensors-23-00144-t001:** Statistical information of the PV sites in Korea.

Location	# of PVs	Statistical Parameter of PV Data from Each Location			
		Maximum	Mean	Conditional Mean	Standard Deviation
Gangwon	72	0.9439	0.1418	0.4334	0.2377
Gyeonggi	77	0.4336	0.0697	0.1932	0.1059
Gyeongnam	112	1.3644	0.2189	0.6407	0.3493
Gyeongbuk	147	0.8453	0.1374	0.3798	0.2102
Gwangju	39	0.9845	0.1499	0.4373	0.2324
Daegu	27	1.7818	0.3044	0.8400	0.4550
Daejeon	5	0.8604	0.0595	0.3635	0.1419
Busan	12	0.0475	0.0086	0.0229	0.0129
Seoul	8	1.1770	0.1986	0.5493	0.3030
Sejong	4	3.2630	0.5486	1.5490	0.8344
Ulsan	4	0.8950	0.1454	0.4104	0.2245
Incheon	10	0.7373	0.1287	0.3522	0.1932
Jeonnam	243	0.5957	0.0985	0.2836	0.1544
Jeonbuk	110	0.4478	0.0700	0.2030	0.1113
Chungnam	113	1.8684	0.3018	0.8458	0.4673
Chungbuk	51	1.3418	0.2063	0.5972	0.3247

**Table 2 sensors-23-00144-t002:** Structure of the proposed TransGRU.

Layer	Name	Dimension	Number of Parameters
0	Input	288	-
1	Transformer encoder; Feedforward input	512	147,968
2	Transformer encoder; Feedforward output	288	147,744
3	GRU	64	67,968
4	FC layer 1	256	16,640
5	FC layer 2	256	65,792
6	Output	24	6168

**Table 3 sensors-23-00144-t003:** Hyperparameters of the model optimization.

Hyperparameter	Value
Batch size	18
Learning rate	0.0001
Optimizer	Adam
Epoch	1000
Loss function	MSE

**Table 4 sensors-23-00144-t004:** PV forecasting error of individual PV sites.

Location	Weather Forecast Error (%)	NMAE10 (%)					
		MLP		GRU		TransGRU	
		VT	DT+	VT	DT+	VT	DT+
Gwangju	5	9.47	7.97	10.23	8.85	8.11	**7.49**
	10	9.51	8.04	10.37	9.04	8.59	**7.62**
	15	10.04	8.66	10.92	9.29	8.74	**7.67**
	20	10.98	8.79	11.60	9.74	9.22	**8.27**
Daegu	5	9.48	8.41	10.07	9.02	9.40	**8.09**
	10	10.35	8.79	10.64	9.42	9.46	**8.21**
	15	10.44	8.99	11.25	9.37	9.55	**8.22**
	20	11.14	9.48	12.12	9.94	10.01	**8.79**

The best performance metric for each weather forecast error case is in **bold**.

**Table 5 sensors-23-00144-t005:** PV forecasting error of PV aggregation.

Location	# of PVs	Capacity (MW)	Weather Forecast Error (%)	NMAE10 (%)					
				MLP		GRU		TransGRU	
				VT	DT+	VT	DT+	VT	DT+
Gwangju	39	29.92	5	5.50	4.73	5.47	5.03	5.28	**4.97**
			10	5.64	4.8	5.68	5.21	5.36	**5.0**
			15	6.13	4.81	6.09	5.49	5.59	**5.12**
			20	6.20	5.03	6.07	5.54	6.05	**5.54**
Daegu	27	34.47	5	5.76	5.29	5.66	5.09	6.43	**5.35**
			10	6.14	5.51	6.22	5.98	6.41	**5.34**
			15	6.67	5.88	6.50	5.12	6.44	**5.53**
			20	7.28	5.86	7.06	5.82	7.27	**6.14**

The best performance metric for each weather forecast error case is in **bold**.

## Data Availability

The weather data used in this study are openly available in Open MET Data Portal (https://data.kma.go.kr/cmmn/main.do (accessed on 8 December 2022)) operated by KMA (Korea Meteorological Administration).

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
