# Peer review of "DTTrans: PV Power Forecasting Using Delaunay Triangulation and TransGRU"

_sensors, 2022, doi:10.3390/s23010144_

Round 1

Reviewer 1 Report

The proposed technique is well explained and worth publishing in this journal.

To improve the quality of the paper, a few suggestions are as follows.

1.      The literature review presented here is highly insufficient and generalized. Please improve it using recent papers.

2.      Eqn. 2 is not clear. Please elaborate.

3.      Few variables are not defined. Please correct it.

4.      The picture quality of waveforms should improve.

5.      Few short forms have been used without giving full forms. Please cross-check throughout the paper properly.

6.      To improve the introduction and reference sections, you should follow quality papers. A few suggestions are as follows. doi: 10.1109/TSTE.2019.2891558, doi: 10.1109/TIE.2018.2890497, doi: 10.1109/TCSI.2020.2996775.

7.      Please go through those papers, and include and improve your literature review portion of the paper.

8.      Elaborate discussions of results. Try to point out each waveform using proper justification.

9.      Rewrite the conclusion section in the summarized form.     

Reviewer 2 Report

In this manuscript, the authors use GRU combined with a Transformer to predict the power output of PV. The topic is relevant to the aim of the journal Sensors. However, the current version cannot be accepted for publication. Detailed information and other comments are listed below:

(1) is the term Delaunay Triangulation developed by the authors or adopted from elsewhere, please provide the reference if cited from other research work.

(2) The authors should state why choosing ML method instead of a physical model to make prediction/regression. Are there any obstacles to adopting physical methods?

(3) Why only use GRU? There are many other time series-based deep learning algorithms, and the authors should better illustrate why not use this instead of others.

(4) It would be better to show both training and test results in the same figure to better view overfitting and underfitting issues.

(5) Please summarize the literature in the introduction section as appropriate, which would avoid making your literature reviews like stacking papers and also help the audience better view your review work and better transaction to the next paragraph.

(6) From the authors' introduction and paper review, it is not clear about the gap, the method is a maturely established model Transformer/GRU, and the target problem is already been taken care of in several papers already. The authors should differentiate theirs from the other authors.

(7) There are ample references that can be used in the introduction for the expansion of ML algorithms applied for regression, the authors can take the following reference as examples for successfully implementing ML for regression tasks: Comparing deep learning models for multi energy vectors prediction on multiple types of building; A general model for flow boiling heat transfer in microfin tubes based on a new neural network architecture. Please consider referring to them and summarizing these papers as appropriate.

(8) Some attention needs to be given to the limitations associated with the methods used in the current work.

Reviewer 3 Report

1.       Abstract: Line 2- …scheduling power system’s’.

2.       Line 11: Aggregated is misspelt

3.       Line 12: ‘on’ average not ‘in’ average.

4.       Lines 19-20 – ‘…global total installed PV capacity was 167.8 …’  Give the correct reference link.

5.       Line 20: …. representing ‘a’ 36% ...

6.       Line 21: … by ‘the’ stochastic...

7.       Line 23:  ... of power system’s’.

8.       Line 24:  …many ‘architectures’...? Approaches perhaps?

9.       Line 26: ‘an’ hourly…

10.   Line 32:  ... techniques have been ‘proposed’ or ‘used’.

11.   Line 33:  … since ‘they’ can ‘solve’ nonlinear…

12.   Lines 34 & 35: ‘networks’ not network

13.   Line 37: What is WPD in full?

14.   Line 46: The work In [9] …. What is ST?

15.   Lines 47-48: Lately Graph neural networks (GNN) were …

16.   Line 49:  .. generation, and further … on going [16-18].

17.   Line 55:  ... be installed further, developing a light model... What does light mean? Computationally light?

18.   Line 60: Delaunay is misspelt

19.   Line 61:  ... weather stations of

20.   Line 62: Vertices of ‘the’ Delaunay…

21.   Line 65-66: … because it transforms noisy input data to more useful time domain features…. Reference?

22.   Line 83: ‘nationwide’ – mention the country here

23.   Line 84: ‘… 7−15% improvement…’. Improvement relative to what?

24.   Figure 1: Give the meaning of FC

25.   Line104: … performance ‘for the’ test set…

26.   Line 158: Do the circles in Equation 5 indicate multiplication? Use solid circles in that case, for clarity

27.   Line 159: ‘… W, U and ??? are…’. And what?

28.   Line 160: Remove the ‘ . ‘ after vector

29.   Figure 6: Use dots to show interconnection points for the different lines/parameters. Also the relation between Fig. 6 and Equations (5) is not clear. The σ terms in Fig. 6 have no subscripts, and where are the br and bz terms? Redraw Fig. 6 better.

30.   Lines 170-174: Give a summary of the general statistics of the 1034 PV plants – e.g. average, minimum, maximum, standard deviation of the plant sizes, number of plants in various plant size ranges…

31.   Line 178: What is the interval of the ASOS weather data?

32.   Line 180: Cleaning is misspelt

33.   Line 181: ‘... sampled every hour.’  Sampled how, or on what basis? Using the hourly mean?

34.   Line 184-185: Why the choice of 96 and 288?

35.   Line 198: Give MLP in full

36.   Line 221-222: Why did you not use 5%, 10%, 15% and 20% of σ rather than 5%, 10%, 15% and 20% of the maximum value of each factor? What is the scientific basis for the latter choice? Is that typical of Korea's weather variations? Could you for example use information spanning longer periods e.g. from ASOS or other weather data sources to come up with the sensitivity factors for σ?

37.   Line 236: `... are located in the way…’. Unclear what is meant by in the way

38.   Line 240: ‘... ϵ1 = 1.3 and ϵ2 = 2...’ What is the basis for these choices of ϵ1 and ϵ2?

39.   Figure 8: Could Figure 8 results be due to location-specific parameters? Show e.g. on the map of Korea and in Figure 2 where these 2 sites (Daegu and Gwangju) are located. Include corresponding data for some other regions aside from just these 2 to show how consistently your method outperforms the other 2 methods.

40.   Line 251: ‘selected weather stations A, B, and C.’ Any basis for choosing specifically these 3 stations?

41.   Figure 9: Increase the horizontal axis length - the axis values’ font size is too small

42.   Line 254: Explain more how the inference is made

43.   Line 255:  ... at ‘a’ certain time step.

44.   Line 258-260: Show the location of these 3 weather stations and the target PV site in a figure.

45.   Line 272-273: How do you simulate limited data? By having e.g. a smaller training set? What is the effect e.g. of reducing the training set proportion?

46.   Figure 10: Why is the TransGRU gap more significant in Fig. 10 a for 15% weather forecast error than the other 3 cases of 5%, 10% and 20% error?

47.   Line 278-279: ‘For example, DT+ shows 19.9%, 16.0%, and 10.3% improvement compared to VT in 20% weather forecast error for Gwangju.’ Unclear what you are referring to here, and which figures you are comparing

48.   Line 202: Is this 30-45% improvement for all NMAE10% values in Table 4 relative to those in Table 3?

49.   Line 284: Provide some additional detail on the concept of VPPs

50.   Line 320: ‘…as weather forecast error increases’. Unclear, rephrase

Round 2

Reviewer 2 Report

The authors resolved all the concerns of the reviewer.